# Qualitative evaluation of rapid implementation of remote blood pressure self-monitoring in pregnancy during Covid-19

**Charlotte Paterson** [1]*, **Elaine Jack**[2], **Brian McKinstry**[3], **Sonia Whyte**[4], **Fiona C. Denison**[3], **Helen Cheyne**[1]

**1** Nursing Midwifery and Allied Health Professions Research Unit, University of Stirling, Stirling, United Kingdom, **2** MRC Centre for Reproductive Health, Queen's Medical Research Institute, University of Edinburgh, Edinburgh, United Kingdom, **3** Usher Institute, University of Edinburgh, Edinburgh, United Kingdom, **4** Liverpool Clinical Trials Centre, University of Liverpool, Liverpool, United Kingdom

☯ These authors contributed equally to this work.
* charlotte.paterson100@gmail.com

**Data Availability Statement:** All data cannot be shared publicly because of ethical restrictions (informed consent was sought for quotes not full transcripts to be published), as approved by NHS

## Abstract

In March 2020, the World Health Organisation named the severe acute respiratory syndrome coronavirus 2 (Sars-CoV-2), which causes corona virus disease 2019 (COVID –19), as a pandemic. Pregnant women were considered at increased risk of developing severe COVID-19 after viral infection. In response maternity services reduced face-to-face consultations with high-risk pregnant women by supplying blood pressure monitors for supported self-monitoring. This paper explores the experiences of patients and clinicians of the rapid roll-out of supported self-monitoring programme in Scotland during the first and second wave of the COVID-19 pandemic. We conducted semi-structured telephone interviews with high-risk women and healthcare professionals who were using supported self-monitoring of blood pressure (BP) In four case studies during the COVID-19 pandemic. 20 women, 15 midwives and 4 obstetricians took part in the interviews. Interviews with healthcare professionals showed that while implementation occurred at pace and at scale across the National Health Service (NHS) in Scotland, implementation differed locally, resulting in mixed experiences. Study Participants observed several barriers and facilitators to implementation. Women value the simplicity of use and convenience of the digital communications platforms while health professionals were more interested in their impact on reducing workload for both women and health professionals largely found self-monitoring acceptable, with only a few exceptions. These results show that rapid change can occur in the NHS at a national level when there is a shared motivation. While self-monitoring is acceptable to most women, decisions regarding self-monitoring should be made jointly and on an individual basis.

## Introduction

Raised blood pressure (BP) affects approximately 20% of pregnancies [1]. Characterised by high blood pressure, pre-eclampsia affects at least 5–8% all pregnancies [2, 3]. Globally, around

Research and Development departments and the Caldicott Guardian of the sponsor NHS Boards. Relevant data have been provided in the manuscript and in supplementary materials. All study findings can be replicated using this data.

**Funding:** The project was funded by the Scottish Government Health and Social Care Directorate (https://www.sehd.scot.nhs.uk/aboutus.html). Fiona Dennison and Helen Cheyne were the grant holders. There was no grant number as this was commissioned work not competitive grant funding. The funder had no role in study design, data collection and analysis, decision to publish, or preparation of the manuscript.

**Competing interests:** BM has a paid consultancy with the Scottish Government to provide advice on the implementation of remote health monitoring. This does not alter our adherence to PLOS ONE policies on sharing data and materials.

**Abbreviations:** BP, Blood pressure; COVID –19, corona virus disease 2019; DAU, Day assessment unit; HB, National Health Service health board; MS Teams, Microsoft Teams; NHS, National Health Service; NICE, National Institute for Health and Care Excellence; RCOG, Royal College of Obstetricians and Gynaecologists; RCT, Randomised controlled trial.

15% of maternal mortality is due to pre-eclampsia so early detection and prevention are paramount [4].

To detect progressive hypertension, and therefore prevent preeclampsia, pregnant women's BP is monitored by health care professionals during face-to-face antenatal appointments. This is the main diagnostic method for detecting preeclampsia [2, 5], however, the recent outbreak of coronavirus and the risk of disease transmission to a vulnerable group (Sars-CoV-2) has presented challenges for face-to-face consultations.

In March 2020, the World Health Organisation designated the severe acute respiratory syndrome coronavirus 2 (Sars-CoV-2), which causes corona virus disease 2019 (COVID –19), as a pandemic. To try and slow growth of the pandemic, the UK government instigated a nationwide lockdown. Individuals identified as being clinically vulnerable or extremely vulnerable, including pregnant women, were recommended to adopt additional protective measures, such as limiting social contact, in order to reduce the risk of developing severe COVID-19. The Royal College of Obstetricians and Gynaecologists (RCOG) issued UK wide recommendations that face-to-face consultations should be limited for high-risk pregnant women to reduce their risk of viral exposure [6]. This required the national health service (NHS) maternity services to urgently find ways to reduce usual face-to-face contacts for women without compromising safe care, particularly for women 'at-risk' of developing preeclampsia.

Supported, remote self-monitoring of BP can either be used to replace BP measurements on the day of usual scheduled clinics (i.e., intermittently) or can be undertaken more frequently and routinely (e.g., daily or weekly) in addition to usual care. When implemented in non-pregnant populations in diverse care settings there is good evidence for its effectiveness in lowering BP, and its acceptability among patients [7, 8]. Further, in a recent Scottish Technology Enabled Care Project, 'Scale-Up BP', telemonitoring reduced face-to-face appointments by 19% for non-pregnant patients with hypertension [9]. Despite apparent efficacy and acceptability, evidence among health professionals has been mixed. Telemonitoring has not been widely adopted by clinicians, largely due to concerns about workload and patients ability to self-monitor [7, 10].

Implementing new models of care at scale is challenging [11, 12] and many factors can influence the success of implementation, e.g. how health care professionals interact with the new model, perceived usefulness of the new model, ease of use, access and time for training and encouragement from senior staff [13, 14]. Although research in general populations has provided evidence of promising outcomes and some acceptability, national implementation of a new model of care in maternity services during a global pandemic is likely to encounter additional and important challenges. To understand the unique challenges of implementing a new maternity service model at scale during a global pandemic we explored the experiences of patients and healthcare professionals of the rapid implementation of a supported, BP self monitoring programme for high-risk and shielded pregnant women in Scotland during the first and second wave of the COVID-19 pandemic. This study provides new insights into issues needed to be addressed to implement BP self-monitoring in routine practice.

### Aims

1. To explore the way in which the supported self-monitoring programme was implemented across contrasting sites,

2. To assess the acceptability, views and experiences of women participating in the supported self-monitoring of BP programme, and

3. To assess the views and experiences of staff involved in the supported self-monitoring programme, including perceptions of barriers and facilitators of successful implementation.

## Methods

### Design

This was a qualitative case series study, using semi-structured telephone interviews. As this study was considered to be a clinical evaluation NHS ethical approval was not required. Instead, the study was approved by appropriate NHS Research and Development departments, the Caldicott Guardian of the sponsor NHS Board and the relevant University of Stirling Ethics Committee. The study is reported in line with Standards for Reporting Qualitative Research (SRQE) [15].

### Setting and sample

In Scotland, universal maternity services are provided through 14 geographical NHS health boards, with oversight and strategic direction provided by Scottish Government Health and Social Care Directorates, and guidance provided by NHS National Services Scotland. This structure enabled a multi-professional working group to be established in March 2020 to co-ordinate the implementation of remote consultation and monitoring in maternity care, and to develop clinical and technical guidance to support remote self-monitoring of BP and urine analysis for high-risk pregnant women. This was available in paper and online form [16]. The Scottish Government also purchased 5000 blood pressure monitors which were distributed to the 14 NHS health boards in May 2020 to enable women to undertake supported remote self-monitoring of their BP. Consistent with UK wide guidelines issued by the RCOG, this national oversight was intended to ensure that the programme was implemented consistently across Scotland. Some tailoring to local NHS health board needs was anticipated, for example, different digital platforms were in use in different parts of Scotland for communications between healthcare professionals and women.

Three NHS health boards were recruited as case study sites. This number of sites was judged to provide a reasonable range of health board characteristics within the limited time available for the study. Sites were selected to represent a range of geographical and clinical contexts [17]. In NHS health board one (HB1) 58% of the population lived in large urban areas. Maternity services were provided by involved one tertiary referral centre and one district general hospital. In NHS health board two (HB2) 39% of the population lived in large urban areas and 40% in semi-urban communities. Maternity services involved one district general consultant unit. In NHS health board three (HB3), over 50% of population lived in rural or remote rural areas and 26% in remote small towns, with maternity services being provided via one district general hospital and 10 community midwife units. Within HB3, organisation and management of maternity services were divided between the north (HB3 north) and south (HB3 south) of the health board due to the geographical size of the board. All health boards had community midwifery services.

The anticipated sample was 15–20 women (5–7 from each site), up to 15 midwives (5 per site) and 10 obstetricians (3–4 per site). This sample size was considered to be sufficient to provide experiences from a variety of women and healthcare professionals. A sample of women were selected and approached using a sampling frame to maximise diversity. Characteristics of women used in the sampling framework included socioeconomic background, age and parity. Women were chosen to provide an even spread of characteristics.

**Table 1. Programme eligibility criteria for women by group.**

| Group | Group 1 | Group 2 | Group 3 |
|---|---|---|---|
| Description | 'High risk' of hypertensive complication | 'Increased risk' of developing pre-eclampsia | Other |
| Relevant conditions | Chronic hypertension | Hypertensive disease during a previous pregnancy | Type 1/ Type 2 Diabetes |
| | Current gestational hypertension | Chronic Kidney disease | Multiple pregnancy |
| | Current pre-eclampsia | Autoimmune disease | |
| | Cystic fibrosis | | |
| | Solid organ transplant | | |
| | Cardiac Conditions | | |

### Eligibility criteria

Pregnant women in the three NHS health boards were eligible to participate in the evaluation if they were taking part in the supported home BP monitoring programme. Three groups of women were eligible for the self BP monitoring programme (see Table 1). To participate in the evaluation, women also had to speak English or have access to an interpreter. healthcare professionals (midwives and obstetricians) who provided antenatal care to women and who had experience of the supported self-monitoring programme were eligible for the evaluation.

### Recruitment

Women were recruited by a member of the local clinical care team, who gave eligible women a 'consent to contact' form, explaining the aims of the evaluation and asking for if their personal details (e.g., postcode, age, parity, email and telephone) could be passed to the research team. If permission was given, they signed the form and contact details were recorded by the local clinical lead for the supported self-monitoring programme and passed securely to the researcher. The researcher then chose women to contacted, based on the sampling framework. Healthcare professionals involved in the programme were initially identified by the local staff leading the implementation of the remote BP self-monitoring. Relevant staff were then sent a participant information sheet by email. A telephone call was arranged between the researcher and potential participants (women and healthcare professionals) to discuss the evaluation and arrange a suitable time for interview if permission was given. Oral consent to participate was obtained using a predefined script which was recorded at the beginning of each interview.

### Data collection

Telephone interviews were conducted and audio recorded with women, midwives and obstetricians from three NHS health boards between August 2020 and December 2020. All interviews were conducted by one researcher (CP). Two semi-structured topic guides were used. Research team meetings were held fortnightly, where interview progress was discussed. Interview topic guides were refined iteratively in response to the initial interviews following discussion in research team meetings, e.g. prompt questions were added or reworded.

For women, prompts included:

- How confident do you feel in using your home BP kit?

- How well was the process explained to you?

- How do you feel about monitoring your own BP?

- Do you feel that you know enough about how to recognise normal/abnormal BP?

Prompt questions added for women in response to team meetings:

- Do you know who/how to contact someone if you have any concerns?

- Do you feel that your midwife/ doctor are available to support you if you need this?

    For staff, prompts included:

- How confident are you in teaching women to use the BP kits?

- What is your experience of using home BP monitoring with women in your care?

- Do you have any particular concerns?

- Do you feel there any benefits/ risks?

    Prompt questions for staff that were added in response to initial interviews:

- What infrastructure do you feel needs to be in place to ensure implementation is successful?

- How does home BP affect the normal care pathway for women in your care?

## Analysis

Participants were anonymised and assigned a code which is used to refer to participants in the results, e.g. S1. Qualitative data analysis software (NVivo version 12) [18] was used to analyse interview audio recordings. All relevant sections of the audio recordings were transcribed and subsequently coded by one researcher (CP). A Framework Approach [19] was initially used to code the data. The analysis framework included the following overarching themes: outcomes (including clinical, service and psychosocial), and barriers and facilitators to implementation. Thematic analysis [20] was then used to identify common patterns in the data under each overarching theme. After coding of the first three interviews was complete, two researchers (CP and HC) met to examine, discuss and refine all themes and subthemes. The remaining interviews were then analysed using the process described above. After all coding was complete, tables were developed in Microsoft Word to create a matrix into which themes and associated data were charted by participant type (i.e., healthcare professional and women) and health board. Opposing and similar views between healthcare professionals and health boards were explored. Preliminary results were shared during a stakeholder webinar to check relatability and accuracy of the findings. No changes were suggested by stakeholders.

## Reflexivity

Research team members in close contact with the data were experienced in health services research, with a range of expertise. CP is a white, female, post-doctoral research fellow with a background in systematic reviewing, implementation science, mental health service evaluation and a clinical background in psychology. She has no clinical experience of maternity services. HC is a white, female, midwife with extensive experience of leading trials and evaluations in maternity services. CP had no relationship with the participants prior to the study. As a senior midwife academic in Scotland HC had worked with some participants in a professional capacity.

## Results

### Participants

Interviews were conducted with 20 women, 15 midwives and 4 obstetricians across the three NHS health boards (see Table 2).

**Table 2. Number of women and healthcare professionals recruited in each NHS health board.**

| Case study site | | Women | Midwives | Obstetricians |
|---|---|---|---|---|
| HB1 | | 8 | 5 | 2 |
| HB2 | | 7 | 5 | 2 |
| HB3 | HB3 North | 3 | 3 | 0 |
| | HB3 South | 2 | 2 | 0 |
| Total | | 20 | 15 | 4 |

NHS health board (HB)

## Implementation in each site

A description of the programme implementation in each NHS health board is presented in Table 3. HB3 has been divided into HB3 North and HB3 South. HB3 covers a large area of Scotland with multiple services, and service design varies. In HB3 North, maternity services are run by midwife- and obstetrician-led maternity teams, while in the rural midwife was supported by locally based midwife- and obstetrician-led maternity teams, whereas in HB3 South, the maternity service is midwife-led, with support from obstetricians from a different health board, not participating in the telemonitoring rollout. It became clear during interviews that these health board characteristics had had an impact on the implementation of remote BP self-monitoring and the associated experiences of women and staff and were therefore analysed separately.

## Outcomes of remote BP self-monitoring

A variety of perceived outcomes of remote BP self-monitoring were identified by healthcare professionals and women. Perceived benefits are organised as clinical outcomes, service outcomes, psychosocial outcomes.

**Clinical outcomes.**   Healthcare professionals and women reported a range of clinical outcomes associated with remote BP self-monitoring. In HB1 and HB2, clinicians reported that implementation led to more women being identified as 'at risk' earlier in their pregnancy, i.e. in the first half of pregnancy, and that some women subsequently started medication earlier. Although one midwife was concerned that women may become overmedicated, other midwives, obstetricians and women saw earlier identification and treatment as a positive clinical outcome and a major benefit of self-monitoring.

> 'Yeah, so I think ultimately *I* got the same treatment but probably with less visits and it probably highlighted things a bit quicker than waiting for reviews'
>
> (S11; woman);

> 'I suppose if we're seeing a subtle change with blood pressure in the early part of pregnancy, it's whether you get treatment on board sooner rather than later which is also good, so I think that's a benefit.'
>
> (S17; healthcare professional).

Another clinical benefit of self-monitoring was that it could be used as 'an evaluation tool' (S35; healthcare professional) to differentiate between women who had genuine hypertension/pre-eclampsia and those that had 'white coat' syndrome. Self-monitoring therefore helped guide appropriate treatment pathways. Additionally, women reported that self-monitoring

**Table 3. Table describing implementation in each NHS health board.**

| Implementation characteristics | NHSB1 | NHSB2 | NHSB3 | |
|---|---|---|---|---|
| | | | **NHSB3 North** | **NHSB3 South** |
| **What service(s) was telemonitoring out in?** | Central maternity DAU providing close monitoring for high risk women | Central maternity DAU, providing close monitoring for high risk women, and community maternity teams | Rural midwife-led and midwife/obstetrician-led maternity teams | Rural midwife-led maternity teams with obstetrician input from a different health board |
| **Who led the implementation?** | Two obstetricians with support from a research midwife | Consultant midwife | Two midwife team leads | One consultant midwife with support from the midwife leading digital health |
| **Who was the local champion?** | One DAU midwife | One DAU midwife and one midwife in each community maternity team | One midwife in each team | One midwife in each team |
| **What training was given to staff?** | Information on the Scottish Perinatal Website[a]; a 'sit down' with lead obstetrician; training delivered by the research midwife including a presentation on identifying eligible women and how to use Florence[b]; written guidance on how to manage medication. During the study period, more detailed guidance on responding to and managing women was developed. Health Improvement Scotland facilitated one shared learning session or all health boards during the rollout. | Information on the Scottish Perinatal Website[a]; training provided by the consultant midwife and technology team including a presentation via MS Teams[c], a local training package and continual on call support. Training detailed (i) how to use Florence and home monitors, (ii) locally developed protocols on eligibility and how to interpret and respond to readings, and (iii) test running Florence as a patient. Health Improvement Scotland facilitated one shared learning session or all health boards during the rollout. | Information on the Scottish Perinatal Website[a]; one lead midwife developed and delivered training via MS Teams[c]. Training covered (i) guidance on how to use machines, and (ii) 'what to do, when to do it, what do you do it something's not right, the implementation of it.'(S49). Health Improvement Scotland facilitated one shared learning session or all health boards during the rollout. | Information on the Scottish Perinatal Website[a]; the digital midwife developed and provided training via MS Teams[c]. Training covered how to identify eligible women and how to manage self-monitoring women. Health Improvement Scotland facilitated one shared learning session or all health boards during the rollout. |
| **Who received training?** | DAU midwives had access to materials | DAU midwives received training. After rollout began, a cohort of trainee obstetricians received training and one community midwife in each team disseminated information throughout team. | Midwife team leads across NHSB3 North and some obstetricians received training. Team leads disseminated information throughout team after rollout began | Midwives across NHSB3 South received training. |
| **How did women access self-monitoring?** | DAU, community teams, GPs | Initially via DAU, then via DAU and community teams | Routine maternity appointments | Routine maternity appointments |
| **Methods for recording and communicating BP** | Women used Florence to record and communicate their BP. Midwives received recordings via Florence and transferred readings to TRAK[d] | Women used Florence to record and communicate their BP. Midwives received recordings via Florence and transferred readings to BadgerNet[e]. | Women recorded and communicated their BP via 15-minute appointments on the BadgerNet application, via weekly telephone appointments with midwives or via a text message including a photo of the machine reading. | Some women tried to record and communicate their BP using Florence, however, there were network and connection issues. Other women used weekly NearMe[f] appointment with their midwives, text messages to named midwife and email to the team email address. |

*(Continued)*

**Table 3.** (Continued)

| Implementation characteristics | NHSB1 | NHSB2 | NHSB3 | |
| --- | --- | --- | --- | --- |
| | | | NHSB3 North | NHSB3 South |
| **What were the local processes for managing women who were self-monitoring (e.g., provide training and information, review readings, receive phone calls, etc.)?** | One DAU midwife managed self-monitoring women two days a week. This included transferring readings from Florence to TRAK, phoning women who had not submitted readings, setting new women up with Florence, and taking Florence related phone calls from women and staff. Emails from Florence were sent to the team DAU account when readings were not submitted. Guidance from hospital obstetricians was sought when needed, e.g., regarding medication commencement. | Named midwives reviewed readings and transferred them onto BadgerNet. Emails from Florence were sent to personal accounts when readings not submitted. A buddy system was to cover annual leave. The DAU used hospital computers to manage women who were already on their caseload. Community midwives used remote laptops to manage home monitoring women who were on their caseload, unless a visit to the DAU was required. Obstetricians in DAU provided advice regarding abnormal BPs when needed. | Named midwives reviewed readings when required. Some midwives had laptops for remote working. Arrangements were made for annual leave cover. Obstetricians either in the local service or nearby services provided guidance when needed. | One midwife reviewed readings once weekly. Women were organised so they communicated readings on the same day weekly. Midwives had laptops for remote working. Input from obstetricians from a different NHS health board was sought, when needed. |
| **Methods and arrangements for contact with women** | Telephone appointments were conducted two weeks after commencing self-monitoring to check in. Women were told by midwives to phone their midwife or triage if readings were abnormal. Women were also prompted to do so by text messages from Florence when readings were abnormal. | Some community midwives text women to 'check in' weekly. Women were told by midwives to phone their midwife or triage if readings were abnormal. Women were also prompted to do so by text messages from Florence when readings were abnormal. | Individual plans for contact with the named midwife via NearME and telephone. Women told by midwives to phone their midwife or triage if readings were abnormal. | Individual plans for contact with the named midwife via NearME and telephone. Women told by midwives to phone their midwife or triage if readings were abnormal. |
| **Approach to abnormal BP parameters** | Midwives and obstetricians linked to this project were aware of and followed parameters set in guidelines [2]. Obstetricians not linked to this project used various parameters | Each woman was given 'sticker' with personalised abnormal BP parameters. | Guidelines were mostly followed. Personalised abnormal parameters were given to women who had particularly low BP or had existing hypertension. | Guidelines were followed, however, please note that few women recruited to home monitor were 'high risk'. |
| **Retrieving home monitors** | Plan to retrieve machines via community midwives during last post-natal appointment or by GP. Administration took on role to follow-up missing machines which was previously done by DAU midwives. | No standardised process. Women tended to drop off machines. | Monitors returned at discharge where possible. No formal process for retrieving machines being used post-natal. | Informal agreement in place that women would return monitors to names midwife at last post-natal check. |

Blood Pressure (BP); Day Assessment Unit (DAU); General Practitioner (GP); Microsoft Teams (MS Teams); National Health Service (NHS); National Health Service Board (NHSB).

[a]. Scottish Perinatal Website [16]

[b]. Florence: digital platform supporting one way communication of self-monitoring results from women to service via text messaging, with automated feedback.

[c]. MS Teams: digital platform supporting video conferencing.

[d]. TRAK: digital platform supporting electronic maternity records.

[e]. BadgerNet: digital platform supporting electronic maternity records and communication between women and staff via the smartphone application.

[f]. NearMe: digital communication platform supporting video calls.

had reduced their anxiety about going into hospital, particularly those with white coat syndrome (whereby a patient's feeling of anxiety in a medical environment results in an abnormally high reading when their BP is measured), therefore potentially reducing the need for medication or admission to hospital.

Another clinical benefit of remote self-monitoring reported by healthcare professionals was that women had more autonomy, independence and control with regards to their BP. As such, women reported being more in tune with their own health.

'*It just feels like, if you can do it at home, it makes you think a bit more about your own health, which I think is a good thing.*'

(S25; woman).

**Service outcomes.** The main service outcome reported by healthcare professionals was changes in workload. All staff in HB1 said that workload had increased. In HB1, only midwives in the day assessment unit were trained to use remote BP self-monitoring, however, women were referred from community maternity services and general practitioners (GPs) (see Table 3). As such, women who usually had BP reviewed and managed in the community instead had BP reviewed and managed in the DAU, therefore increasing the caseload and workload of the DAU:

'*We have one main desk with one phone and it's constantly going with either midwives referring people for the home blood pressure monitoring to be setup on it, or that they're seeing someone who's already on the monitoring and their blood pressure is high, or women just phoning in with high blood pressures*'

(S16; healthcare professional).

In all other health boards, clinicians reported that workload reduced because there were fewer face-to-face contacts and less travel to see women. In HB2, which is similar to HB1 in that it is a central 'high risk' maternity service, BP self-monitors were administered by DAU and community midwives. Care for women in the community who were self-monitoring therefore stayed in the community which avoided overloading the central 'high risk' maternity service and was seen as a benefit for women:

'. . .*they're slightly on a red pathway if they're on this [BP self] monitoring, but it keeps them at home and it lets their community staff see them more and their community staff can have much more input instead of coming in*. . .'

(S09; healthcare professional).

Interestingly, while speculating on the local approach to self-monitoring implementation, an obstetrician in HB1 said that:

'. . .*what we set up [for the self-monitoring service] was in the day assessment unit as opposed to in the community*. . . *that has led to a little bit of duplication of work because the assessment unit is also doing telephone follow-ups for the women, while they're still having their regular [community] midwife checks*'

(S13; healthcare professional).

The obstetrician proposed that:

'*if it was more community based I would suspect that we would see workload to the day assessment unit would go down.*'

(S13; healthcare professional).

Clinicians in HB1 and HB2 also reported different approaches to NICE guidelines [2] about abnormal BP parameters that require close monitoring and treatment. HB1 'strictly'(S05; healthcare professional) followed new guidelines, resulting in frequent phone calls and sometimes face-to-face visits from some women, i.e. women who had borderline treatment level or had 'high normal' BP:

'*people that were sort of borderline treatment level, but weren't quite treatment level, we knew they were gonna be a lot of work because they kept phoning back because, as per Florence* [the telemonitoring system] *they were told to.*'

(S52; healthcare professional).

In contrast, healthcare professionals in HB2 said that they personalised BP parameters for women who had 'high normal' BPs. As such, there was a higher threshold for asking women to contact the service via telephone or face-to-face. Despite this, it should also be noted that women with 'high normal' BPs were also highlighted by midwives in HB2, HB3 North and HB South as potentially increasing workload, but that their services had few women using remote BP self-monitors.

Another service outcome reported by some staff (HB2 and HB1) was that self-monitoring discouraged 'unnecessary' or 'unscheduled'(S17; healthcare professional) tests. These clinicians described that checking additional parameters during face-to-face appointments was not uncommon, although such tests were not driven by clinical guidelines. Self-monitoring was therefore viewed as having reduced these unnecessary tests.

**Psychosocial outcomes.**    Many midwives and obstetricians across health boards described feeling reassured by women self-monitoring their BP. This was the case, for example, where women with was borderline treatment level BP were sent home without treatment. This was particularly ture in HB3 North and South where women lived a significant distance from a health centre or hospital. Similarly, self-monitoring provided reassurance for midwives in confirming that women with white-coat syndrome did not need further monitoring or treatment.

'Well, I'd say for the likes of the woman with the white coat hypertension, so I know it's going to be higher for her when she's in there, but when she's at home I know her BP is fine, so it's removing that sort of worry of 'has she really got preeclampsia or essential hypertension or anything like that' because we know that is it white coat hypertension because she's totally fine when she's at home.'

(S21; healthcare professional).

However, *h*ealthcare professionals also described a range of factors which led to sense of unease, e.g. concern that important symptoms (e.g. oedema, headache, etc.) were not routinely assessed as well as BP or protein in urine.

'I guess that when they're inputting their blood pressure it might be normal but actually they've got increased swelling and a headache and dizziness. So I feel like it *[telemonitoring]*

is good in monitoring that specific part but it's not good in the sense that normally these girls would be coming in for us to see and we could visually see them.'

(S08; healthcare professional).

Midwives described the importance of women using the telemonitoring system properly and concern that some women may deliberately submit low readings to avoid admission to hospital.

*'it's. . . making sure that the women understand how to use things, how to record things, and how to action things so that there's absolutely no risk of someone sitting there with a result that you would want acted on for example and it's not connected in, especially when it's not a wee walk round the corner to the midwife.'*

(S41; healthcare professional);

*'I think they* [women] *may, eh, sort of, eh, put in a lower reading than what it might actually be so that they don't have to go to hospital. That is one of the concerns that we actually had.'*

(S45; healthcare professional).

Technology issues were also identified as a source of concern for midwives across all health boards. Some midwives worried about inaccurate readings and had a lack of trust in the reliability of the machines because some home monitors had given '*wildly different readings*'(S49; healthcare professional), or that they were extremely high compared to the midwives' readings. Midwives also expressed fear that they may miss women's results when submitted to an electronic system (e.g. Florence or BadgerNet (see Table 3)) or through text, e.g. because annual leave cover had not been organised or mobile signal was poor. However, it was also acknowledged by healthcare professionals that they would get used to this new way of caring for women in time, but it may take longer for some:

*'I suppose it's just a different way of delivering the service and keeping women safe from a hypertension point of view.'*

(S17; healthcare professional);

*'for. . .midwives who are not used to working with technology so much, it's been, it's maybe been more of an adjustment.'*

(S35; healthcare professional).

For most women, supported self-monitoring was considered easy to use and reassuring because they were able to keep track of their health between appointments, and it was described as a '*fabulous*'(S10; woman) and '*really positive*'(S30; woman) experience. Most women reported feeling confident about using the guiding information provided for judging their BP level. Some women even commented that they would want to use it again in future pregnancies, that they would like to have used it earlier in their current pregnancy, or that they wish they had use it in previous pregnancies.

Yeah it [self-monitoring] was good. It was easy to use, the introduction to it was good and I got quite regular phone advice when I needed it

(S11; woman);

*'I think the full idea of having those available for a patient to be able to monitor their own blood pressure at home is an amazing idea. I wish they'd had it previously with my other pregnancies'*

(S31; woman).

Five women reported anxieties relating to self-monitoring. Two women were concerned about their health which resulted in excessive self-monitoring. One woman described stressful personal events increasing her BP readings, which in turn lead to more stress. Another woman was concerned about the personal responsibility of self-monitoring and felt uncertain that she was using the monitor correctly. Despite some concerns, of these women, only two reported *'prefer[ing] a professional to do it [monitor their BP]'*(S24; woman).

Another important psychosocial outcome reported by women was having to travel to fewer appointments. This benefited women because it reduced childcare issues, increased flexibility for those that worked and it saved women time, especially for those living remotely in HB3 North and HB3 South. Women appreciated the flexibility of choosing the time to submit their BP readings, particularly when then were still working or had busy lives.

### Barriers and facilitators to implementation

**Suitability of women.** All midwives and obstetricians identified various categories of women that were considered unsuitable to home monitor. This included those who were not likely to take responsibility for their own health, who did not understand the instructions (e.g., due to learning disabilities), who received complex care (e.g., social work was involved), who had illiteracy or language barriers, who were very young (i.e., <16 years old), who were homeless or in a refuge centre, or who were vulnerable in any other way. Midwives also discussed the suitability of self-monitoring for women who were anxious:

*'I would say that the benefits could also be the risks. So the benefit is that if you have an overly anxious person then they have that peace of mind that they have that machine there that they can press a button and it can tell them that their blood pressure is fine. Equally, if you have that anxious person she could be doing it every 2 minutes, becoming more anxious that it could go up, so you know they're much and much the same.'*

(S45; healthcare professional).

**Support for women.** Most women viewed supported self-monitoring as an addition to their care rather than a replacement for midwives. Where women had uncertainties or concerns, they reported knowing who to phone. Most women noted that they felt a sense of reassurance and support from their midwives because *'you always have people to help at the other end of the phone'*(S38; woman). Some women reported having regular phone advice from their midwives and feeling *'really supported'*(S27; woman).

Midwives and obstetricians described various measures they used to support women while self-monitoring, for example, communicating to women the importance of contacting the service if symptoms other than high BP were experienced and clearing communicating that women were responsible for doing so. HB1 used follow-up telephone appointments to contact women two weeks after receiving the self-monitors to ensure they understood what they were doing, and that they were doing it. For health boards using Florence, reminder text messages were automatically sent to women and automated emails were sent to midwives to notify them that a BP readings had not been submitted, and women were subsequently phoned. Two health

boards used the 'teach-back' method to ensure women understood their responsibilities and to identify unsuitable women, i.e. women were asked to teach back what they had just learned from midwives to verify their understanding.

**Staff buy-in.** All staff saw obstetrician buy-in as essential for successful implementation because, for most women considered to be 'high risk', '*the obstetrician determines what that care plan is going to be*'(S39; healthcare professional). Some obstetricians believed that colleagues and junior trainees had received information about self-monitoring and had responded positively. However, midwives in HB1, HB2 and HB3 South reported that obstetricians, other than those directly connected with the project, either did not know about or did not 'buy-in' to the self-monitoring service. In HB1, midwives thought that a lack of obstetrician buy-in created inconsistencies in care and in HB2 this was perceived as a barrier to midwives promoting the remote self-monitoring service resulting in fewer women using it. In one hospital in HB1, one obstetrician prevented implementation from beginning due to a lack of resources.

> '*the biggest barrier was engagement from the obstetricians, believe it or not, cos I felt as if they'd have been more engaged at the very beginning, em, then they would have been instructing the midwives*'

(S39; healthcare professional).

Midwife buy-in was also seen as essential for successful implementation, however, the extent to which this occurred reportedly varied. Some midwives initially viewed the implementation negatively, but those leading the implementation believed that buy-in increased over time due to the second wave of COVID-19 and a realisation that remote monitoring may be required longer-term. Inversely, other midwives reported seeing the benefits of self-monitoring that were initially 'sold' to them, i.e., that it would reduce workload for midwives and foot fall in hospital. However, over time, due to unmet expectations, i.e., workload increasing rather than decreasing in HB1, some midwives viewed self-monitoring towards the end of the study period as '*just another task that's been added to their role*'(S05; healthcare professional).

**Implementation leader.** The person leading the implementation appeared to be an influential. The implementation leader developed training, local paperwork and protocols, provided support to staff during the implementation, and they promoted the use of home monitors amongst midwives and obstetricians. The implementation leader varied between health boards (see Table 3) and appeared to be most effective when they were a midwife who knew the maternity teams, had experience of implementing new initiatives, was visible to staff during the implementation and had the authority to make decisions regarding local processes for the implementation.

**Staff time and capacity.** Time and capacity were seen as barriers to implementation across all NHS health boards. There were time limitations owing largely to midwife and obstetrician shortages during the COVID-19 pandemic. As such, there was competing demands between new ways of working and maintaining running of usual services:

> '*initially we thought it would be on one morning we'd have both face-to-face and telephone consultations but obviously that meant taking more staff out of the current running of day assessment unit*'

(S10; healthcare professional).

There were concerns across health boards that midwives were being '*saturated*'(S39) or '*bombarded*'(S07) with new information at the beginning of the implementation, e.g. new mandatory training, training for using new methods of holding video consultations (e.g. Near Me), BadgerNet training and learning new ways of working during COVID-19.

Some midwives initially felt that, in that context, self-monitoring was '*quite difficult to deal with*'(S45) because '*it's very time consuming having to do the phone calls*'(S10). In HB1, HB2 and HB3 North, midwives also reported time challenges in developing protocols for responding to and managing women who were self-monitoring, delivering training to all team members or getting the whole team together to discuss news ways of working. Later in the implementation, the added support of bank staff that knew the service well was identified as a facilitator in HB1.

**Training and clinical guidance.** Many midwives and obstetricians appreciated the relevant information that was available across Scotland. Shared learning across health boards, facilitated by Health Improvement Scotland, was seen to benefit implementation, and staff thought the guidance from the Scottish Perinatal Website [16] was clear. As such, all midwives reported feeling confident in teaching women how to use the machines.

There were mixed views between health boards regarding the information, training and guidance provided locally. Midwives in HB2 reported that their localised training and guidance, with involvement from the technology team, was beneficial to the smooth running of the service, however, this was thought to be lacking in HB1 and HB3 North. Standardised procedures for managing women after the initial self-monitoring appointment were unclear, particularly for women with a 'high normal' BPs and for women beginning or changing medication. This lack of clarity led to team members working in different ways and feeling as though they '*just kind of muddled through*'(S44; healthcare professional).

'*I don't think we've had enough support at the beginning of it and a lot of the time, we're not really sure what we're doing. . .I think if they'd had all the guidelines and the exact flowcharts of what to do when A, B, or C happens at the beginning, it would have been much smoother*'

(S44; healthcare professional);

'*I think there's probably a lack of guidance written down as to what to do when it deviates from normal*'

(S49; healthcare professional).

'*[clinical guidance] feels very unclear and I think it's leading to me and my colleagues doing different things. . .*'

(S16; healthcare professional)

In HB1, guidance was being developed during the study period and there was a perception that the organisation of local processes improved over time, which was received well and valued:

'*we've got better processes in place, we've got better files in place, which I've done and are now in date order. Just a bit more organisation of all our documentation to know where we are with things. So in the defence of it all, things have improved*'

(S52; healthcare professional).

**Infrastructure and equipment.** Two main telemonitoring platforms were used during the study: Florence and the Badgernet App. Women liked how easy Florence was to use, while staff found it time consuming because it was not integrated with patient records. Inversely, staff liked the Badgernet App because it synced to patient records, while women found it difficult to use and unintuitive.

Midwives in HB1 also experienced issues finding an available phone and computer in an appropriate location to make confidential phone calls. Further, they reported needing a dual monitor to streamline the process of integrating readings to the patient database and for reviewing and calling women. Unfortunately, this equipment was not available, which was perceived to slow down self-monitoring tasks. Conversely, Community midwives in HB2 and HB3 South each had a laptop and therefore had remote access to NHS databases, which was seen as beneficial to implementation.

Midwives in HB2 and HB3 North reported that the cuffs on the BP monitors were too small for some women, leading to skewed readings. Larger cuffs did not arrive in HB3 North until 6 weeks after the monitors, therefore slowing down recruitment. There were also issues with some home monitors providing unusual readings in HB3 North and South. Mobile networks were also identified as a problem for implementation in rural and island areas. Further, some mobile networks blocked texts from Florence:

> 'some phone networks don't actually allow the texts for Florence to come through. . . someone had that mobile network and when they phoned them they said 'no we won't let any texts from that number come through', not sure why, so they've been having to use their husband's or their partner's phone numbers to be on the system and I don't know how quite appropriate that is. . .'
>
> (S16; healthcare professional).

In rural areas of HB3 North and South, issues with mobile signal and Wi-Fi were major barriers for communication between women and midwives, e.g., women did not receive reminders from Florence or midwives did not receive readings from women. As such, different ways of communicating were used, depending on the woman's location and preference. For example, women submitted their readings via the BadgerNet application or weekly telephone or NearMe appointments with their midwives.

## Discussion

Over the last two years, there has been a need to reduce face-to-face contacts for pregnant and postnatal women in the NHS in the context of COVID-19. Women who are at high-risk of developing hypertensive complications of pregnancy or are shielding due to serious underlying medical conditions and require regular monitoring therefore need to monitor BP and protein in urine at home. In this study we investigated midwive's, obstetrician's and women's experiences and acceptability of supported self-monitoring in three Scottish health boards with four distinct services: HB3 North, HB3 South, HB1 and HB2.

Overall, this study shows that radical change can occur at pace and at national scale within the NHS. Implementing change in the NHS is notoriously difficult. For example, continuity of midwifery care has been shown to give clinical and psychosocial benefits for women and their babies and has been central to UK maternity policy for over 30 years, yet sustained implementation at scale has not yet been achieved [21, 22]. However, based on our research, it appears that the Covid-19 pandemic has acted as a catalyst for change towards digital services at a

national level. It is possible that the pandemic created a shared motivation for innovation that has facilitated digital health implementation on a national level.

Our findings show that both women and healthcare professionals expressed concern about home-self-monitoring, as found in other qualitative investigations of digital support [7, 23]. This theme was stronger within the healthcare professionals in our study. Healthcare professionals were worried that women may not submit BP readings at all, or that they may submit incorrect readings to avoid hospital visits. A number of staff also reported preferring face-to-face appointments to allow for a more holistic assessment of women. In contrast, the vast majority of women reported liking the home BP telemonitoring system due to the convenience and flexibility. Women also described a sense of ownership and increased awareness of their own health, as described in studies of BP self-monitoring in other populations [8, 24]. The contrast between healthcare professionals and patients has also been highlighted in other research in the pregnant population. A discourse analysis of online, offline, and unofficial sources of information around pregnancy and high blood pressure also found a contradiction between the paternalistic discourse of clinicians and the lay discourse of women that both sought control [25]. Interestingly, despite clinicians concerns about passing control to women, few women reported not following advice while self-monitoring in our study, which corresponds to findings about telemonitoring of BP in the general population [26].

Although staff reported some anxieties in relation to women self-monitoring in our study, they also acknowledged that there were benefits of BP telemonitoring. As reported in another study evaluating BP self-monitoring [7], clinicians in our study described that self-monitoring allowed multiple BP measurements to be taken using self-monitors which led to more confident diagnoses and reassurance that women were on appropriate treatment paths faster. Women in our study mirrored this sense of reassurance in knowing they were on the correct treatment path due to more accurate BP measurements, which also reflects other evaluations of BP self-monitoring [7].

A notable difference in healthcare professionals' experiences across health boards in this study was the change in workload. While some services experienced no change or a reduction in workload, a major challenge perceived in one health board was that having women self-monitor was time consuming and increases workload, as found in other BP monitoring evaluations [7, 9]. Starting patients on self-monitors and a lack of integration between the self-monitoring system and electronic patient records was seen as time consuming in other studies, which was also reported by some participants in our study [7, 9]. However, in our study the workload increase was also largely attributed to the telemonitoring service being implemented in the central high risk maternity service, as opposed to in community maternity services. As such, women's BP was managed by the high risk service much earlier in pregnancy. In the high risk service, this resulted managing more women's BP (e.g. calling women and updating electronic records) and more women contacting the service about their BP much earlier in their pregnancy, overloading an already busy service.

Other factors influencing BP telemonitoring implementation in our study include detailed and standardised clinical guidance, formal training, staff buy-in, and involvement of local midwifery leaders. When these factors were realised, they contributed to successful implementation. An earlier review on factors affecting the success of telehealth implementation speaks to many of these factors, as do previous implementation studies and qualitative studies in maternity practice and community nursing align with these factors [12, 27, 28]. Therefore, perhaps unsurprisingly, when these factors were lacking in health boards in our study, they hindered implementation of the self-monitoring service.

## Strengths and limitations

As the study was based on real time implementation in characteristically different health boards, this study provides insights into issues which need to be addressed for BP self-monitoring to be routine practice, in a variety of real world services. Examples include, integrating telemonitoring data with electronic patient records, implementing telemonitoring across services (e.g. community, high risk, and other linked maternity services), additional clinical guidance and involvement of local midwifery leaders, and user friendly platforms for communication. The integration of women's and clinician's views is another major strength of this study, providing multiple perspectives.

A limitation of this study was that recruitment was slow in some sites, leading to a small sample of women for qualitative data collection. We also experienced challenges in recruiting the planned obstetricians sample size from all site, therefore, our study findings lacks this perspective.

Another limitation of this study is that transcriptions were not used during data processing and analysis stages, and one researcher coded most of the data. To assure rigour and ensure findings were relevant and experiences were accurately captured, we conducted credibility checks, involving concept checking with another researcher and in regular study meetings. We also held a stakeholder webinar where findings were presented and feedback was sought.

A final limitation of this study was that quantitative implementation data was not collected and analysed. Quantitative measures of implementation would have provided more insight into workload changes and reach, which would have been beneficial to the full implementation process. Future studies would benefit from using quantitative methods to measure the flow of women through services, women's compliance with guidelines and workload.

## Conclusions

Overall, this research demonstrates that rapid change can occur in the NHS on a national level when there is a shared motivation for change. Implementation varied across study sites and a number of influencing factors were identified which, which should be considered in future implementation strategies for digital health. This study showed that women were almost universally supportive, in comparison to staff, therefore, digital health can be embraced by the NHS without reducing patients' perceived quality of care. In doing so, it is key that midwives are supported to adapt to new ways of working and to use a digital platform that suits both staff and patients. Finally, it is clear that self-monitoring is not appropriate for all women, and deciding who should self-monitor should be a shared decision made on an individual basis.

## Supporting information

**S1 Dataset.**
(DOCX)

## Acknowledgments

Angela Niven, Holly Innis and staff of the Edinburgh Clinical Trials Unit assisted with the set up of the project. Andrew Stoddart and Roz Pollock contributed to study oversight. Thank you to all those involved in facilitating the implementation of the home monitoring service, the women and healthcare professionals who agreed to be interviewed for this evaluation, to all staff involved in supporting the digital communications platforms, and to the staff (particularly Elaine Jack, Mairi Milne, Joan Kelly, Jenny Wilde, Jackie Lambert and Maureen McSherry) and NHS health boards for referring staff and women for interviews.

## Author Contributions

**Conceptualization:** Brian McKinstry, Fiona C. Denison, Helen Cheyne.

**Data curation:** Charlotte Paterson, Elaine Jack.

**Formal analysis:** Charlotte Paterson, Helen Cheyne.

**Funding acquisition:** Fiona C. Denison, Helen Cheyne.

**Investigation:** Charlotte Paterson, Helen Cheyne.

**Methodology:** Brian McKinstry, Helen Cheyne.

**Project administration:** Sonia Whyte.

**Supervision:** Brian McKinstry, Fiona C. Denison, Helen Cheyne.

**Validation:** Charlotte Paterson, Elaine Jack, Brian McKinstry, Helen Cheyne.

**Visualization:** Charlotte Paterson.

**Writing – original draft:** Charlotte Paterson, Fiona C. Denison, Helen Cheyne.

**Writing – review & editing:** Charlotte Paterson, Elaine Jack, Brian McKinstry, Sonia Whyte, Fiona C. Denison, Helen Cheyne.

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
