## [Decision Letter · Decision Letter 0]

22 Apr 2022

PONE-D-21-36755Qualitative evaluation of a rapid rollout of remote blood pressure monitoring in pregnancy during Covid-19PLOS ONE

Dear Dr. Paterson,

Thank you for submitting your manuscript to PLOS ONE. After careful consideration, we feel that it has merit but does not fully meet PLOS ONE’s publication criteria as it currently stands. Therefore, we invite you to submit a revised version of the manuscript that addresses the points raised during the review process.

We look forward to receiving your revised manuscript.

Kind regards,

Dhananjay Singh, Ph.D.

Academic Editor

PLOS ONE

“I have read the journal's policy and the authors of this manuscript have the following competing interests: BM has a paid consultancy with the Scottish Government to provide advice on the implementation of remote health monitoring.”

Additional Editor Comments:

It is not clear what is the novelty of this paper. Please be specific what is your contribution. It is not obvious.

The paper has several flaws that required attentions. Firstly, the justification of tables are confusing and hard to understand.

Although the subject is quite pertinent and result are important, the paper is poorly structured.

Reviewers' comments:

Reviewer's Responses to Questions

**Comments to the Author**

1. Is the manuscript technically sound, and do the data support the conclusions?

Reviewer #1: Partly

Reviewer #2: Yes

2. Has the statistical analysis been performed appropriately and rigorously? 

Reviewer #1: N/A

Reviewer #2: N/A

3. Have the authors made all data underlying the findings in their manuscript fully available?

Reviewer #1: Yes

Reviewer #2: No

4. Is the manuscript presented in an intelligible fashion and written in standard English?

Reviewer #1: Yes

Reviewer #2: Yes

5. Review Comments to the Author

Reviewer #1: It's interesting to study the developing of serve COVID-19 among pregnant women and monitor their health through routine blood pressure measurements and self-monitoring.

The general work and the idea were good but they need some minor changes.

The authors need to demonstrate the abbreviations first time they appear like BP and NHS in the "abstract".

The "introduction" need some literature study to understand the way other people work, their weaknesses and how the authors study overcome them.

The paper need more analysis, most of the results were theoretical which make it some time confusing. The statistics or the analysis tool process need to be included especially in the "data collection" and "analysis" part.

More efforts in the scientific language some paragraphs are hard to understand, the authors need to create clear link between the paragraph and as a suggestion use the active voice, sometimes the passive voice blur the important points.

The authors seems to be neglected some citations and more references need to be added.

The discussion need some conparsion to highlight how important is the study finding.

The limitations could use some future suggestion.

Reviewer #2: The paper is rather comperhensive and well written.

It is not clear how the sample size was determined. The author said the sample is anticipated to provide adequate range of experiences. We have already learned alot from the selected number of sites however it will add value to the work if the author explained more how this size is determined.

I do understand the confedentiality of the data and that they could not make it available to the reader. Nevertheless they could have provided some percentages or so instead of using (most women felt, most women viewed, many, ...etc) in the result section. Doing so made the result section more similar to the discussion while the later resembled a long piece of conclusion.

6. PLOS authors have the option to publish the peer review history of their article (what does this mean?). If published, this will include your full peer review and any attached files.

Reviewer #1: No

Reviewer #2: **Yes: **Rehab Ahmed

---

## [Author Response · Author response to Decision Letter 0]

16 Aug 2022

We have reviewed the reference list and updated it. 

It is not clear what is the novelty of this paper. Please be specific what is your contribution. It is not obvious.

We have edited the introduction to help specify the novelty of this paper. Specifically, we have explicitly stated this study’s contribution: ‘This study provides new insights into issues needed to be addressed to implement BP self-monitoring in routine practice.’

The paper has several flaws that required attentions. Firstly, the justification of tables are confusing and hard to understand.

We have edited the table titles to increase clarity and reduce confusion. 

Although the subject is quite pertinent and result are important, the paper is poorly structured.

We have reworked the introduction with the aim of improving the structure. The order of sections in the Methods have also been edited to provide a better structure, which is consistent with other papers published in PlosOne. Further, the results have been restructured to combine health professionals and women’s views, to highlight the similarities and differences in views of these groups of participants. Some themes have been relocated or merged, where similar points were made across themes, e.g. ‘Communication/Dissemination of Information’ was merged with ‘Training and clinical guidance’, ‘Staff buy-in’ and ‘Implementation leader’ have been moved to ‘Barriers and facilitators to implementation’, ‘Measures to avoid risks’ has been relabelled ‘Support for women’, and subthemes in ‘Psychosocial outcomes’ and ‘Service outcomes’ have been removed. 

The authors need to demonstrate the abbreviations first time they appear like BP and NHS in the "abstract".

Abbreviations have now been described in the abstract.

The "introduction" need some literature study to understand the way other people work, their weaknesses and how the authors study overcome them.

We have introduced additional literature in the introduction and restructured the introduction in order to highlight usual practice before introducing the global pandemic which has instigated supported self-monitoring of BP in Scotland. 

The paper need more analysis, most of the results were theoretical which make it some time confusing. The statistics or the analysis tool process need to be included especially in the "data collection" and "analysis" part.

The ‘Data collection’ and ‘Analysis’ sections have been reworded and additional detail has been provided. However, this paper uses a qualitative research design and did not involve collection of any clinical or other quantitative data, therefore, no statistical analysis was appropriate. Qualitative research methods involve collection of data in the form of words, namely views and opinions of relevant informants. We were unsure if this qualitative data is what the reviewer considered to be ‘theoretical’, however it is the essential nature of qualitative research. Informants' views and opinions may perhaps be considered subjective however careful sampling, expert interviewing procedures and rigorous qualitative data analysis is widely recognised to produce robust and valid evidence. See for example: Pope, C. and Mays, N., 1995. Qualitative research: reaching the parts other methods cannot reach: an introduction to qualitative methods in health and health services research. bmj, 311(6996), pp.42-45. We are confident that qualitative methods were appropriate to address our research aims. 

More efforts in the scientific language some paragraphs are hard to understand, the authors need to create clear link between the paragraph and as a suggestion use the active voice, sometimes the passive voice blur the important points.

We have attempted to use more scientific language and the active voice to increase clarity of important points. 

The authors seems to be neglected some citations and more references need to be added.

We have noticed that some statements required citations and have included these in the introduction. We have also included some additional citations to provide a broader picture of the current literature. 

The discussion need some comparison to highlight how important is the study finding.

We have included additional studies as comparisons in the discussion. 

The limitations could use some future suggestion.

We have now indicated how we believe implementation could be improved in future based on the experience in this study, and we have made suggestions for future research. 

It is not clear how the sample size was determined. The author said the sample is anticipated to provide adequate range of experiences. We have already learned alot from the selected number of sites however it will add value to the work if the author explained more how this size is determined.

Additional detail has been provided with regards to the range of characteristics of the sample of women and the number of case sites recruited.

I do understand the confidentiality of the data and that they could not make it available to the reader. Nevertheless they could have provided some percentages or so instead of using (most women felt, most women viewed, many, ...etc) in the result section. Doing so made the result section more similar to the discussion while the later resembled a long piece of conclusion.

The use of quantification in qualitative research is controversial and we feel that while our study sample is appropriate for our study design use of percentages to quantify themes would be inappropriate and potentially misleading. We refer to Monrouxe & Rees, 2020 (Monrouxe, L.V. and Rees, C.E., 2020. When I say… quantification in qualitative research. Medical education, 54(3), pp.186-187.) The epistemic framework of qualitative research is underpinned by subjectivity and interpretation, use of quantification (e.g use of percentages), may give a false impression of objectivity and implies that the sample is representative. This undermines more complex and nuanced interpretations of the meaning of qualitative data.

---

## [Decision Letter · Decision Letter 1]

11 Nov 2022

Qualitative evaluation of rapid implementation of remote blood pressure self-monitoring in pregnancy during Covid-19

PONE-D-21-36755R1

Dear Dr. Paterson,

We’re pleased to inform you that your manuscript has been judged scientifically suitable for publication and will be formally accepted for publication once it meets all outstanding technical requirements.

Kind regards,

Sandra C. Buttigieg, MD PhD FFPH

Academic Editor

PLOS ONE

Additional Editor Comments (optional):

Reviewers' comments:

Reviewer's Responses to Questions

**Comments to the Author**

1. If the authors have adequately addressed your comments raised in a previous round of review and you feel that this manuscript is now acceptable for publication, you may indicate that here to bypass the “Comments to the Author” section, enter your conflict of interest statement in the “Confidential to Editor” section, and submit your "Accept" recommendation.

Reviewer #2: All comments have been addressed

Reviewer #3: All comments have been addressed

2. Is the manuscript technically sound, and do the data support the conclusions?

Reviewer #2: Yes

Reviewer #3: Yes

3. Has the statistical analysis been performed appropriately and rigorously? 

Reviewer #2: N/A

Reviewer #3: Yes

4. Have the authors made all data underlying the findings in their manuscript fully available?

Reviewer #2: No

Reviewer #3: Yes

5. Is the manuscript presented in an intelligible fashion and written in standard English?

Reviewer #2: Yes

Reviewer #3: Yes

6. Review Comments to the Author

Reviewer #2: The authors addressed my questions one by one, Now I do think this paper is suitable for publication

Reviewer #3: (No Response)

7. PLOS authors have the option to publish the peer review history of their article (what does this mean?). If published, this will include your full peer review and any attached files.

Reviewer #2: **Yes: **Rehab Ahmed

Reviewer #3: **Yes: **Ritu Chauhan

---

## [Editor Report · Acceptance letter]

16 Nov 2022

PONE-D-21-36755R1 

Qualitative evaluation of rapid implementation of remote blood pressure self-monitoring in pregnancy during Covid-19 

Dear Dr. Paterson:

I'm pleased to inform you that your manuscript has been deemed suitable for publication in PLOS ONE. Congratulations! Your manuscript is now with our production department. 

Kind regards, 

on behalf of

Professor Sandra C. Buttigieg 

Academic Editor

PLOS ONE